# Using species distribution models to locate the potential cradles of the allopolyploid *Gypsophila bermejoi* G. López (*Caryophyllaceae*)

**Miguel de Luis, Julio Álvarez-Jiménez, Francisco Javier Rejos, Carmen Bartolomé** [ID]*

Departamento de Ciencias de la Vida, Facultad de Ciencias, Universidad de Alcalá, Alcalá de Henares (Madrid), Spain

* carmen.bartolome@uah.es

**Data Availability Statement:** All relevant data are within the paper and its Supporting Information files.

## Abstract

Polyploidy has been an influential force in plant evolution, playing a crucial role in diversification. Differences in polyploid and diploid distributions have been long noted, with polyploid taxa especially abundant in harsh environments. These plants have higher photosynthetic rates and/or higher tolerance to water deficits. Moreover, there is data pointing to an increase in the rate of unreduced gamete formation by plants under conditions of stress. Accordingly, a higher frequency of polyploid individuals would be expected in populations living under extreme environments, a phenomenon that may be relevant when considering the origin of allopolyploid species. Hybridization between distinct autopolyploids is known to produce allopolyploids and hence, a high frequency of compatible autopolyploids in an area could enhance the formation of stable populations of the corresponding allopolyploid hybrid. Here we consider the allopolyploid species *Gypsophila bermejoi* G. López and its parental taxa *G. struthium* L. subsp. *struthium* and *G. tomentosa* L. We have used Species Distribution Models to locate areas with low bioclimatic suitability for both parental taxa during the Last Glacial Maximum (LGM), hypothesizing that the rate of tetraploid hybrid formation would be higher than expected where low suitability areas of both parental species overlap. We selected those areas taking into account the strict gypsophyllic nature of these taxa. There is data pointing to a post-glacial origin of the current *G. bermejoi* populations and according to our hypothesis, such locations could be centers for hybrid tetraploid formation or potential cradles of this species. Indeed, potential Mid-Holocene cradles were also identified in this manner. The evolution of bioclimatic suitability in both LGM and Mid-Holocene cradles was studied to assess the possible survival of the hybrids, and the current distribution of *G. bermejoi* proved to be consistent with our hypothesis.

## 1 Introduction

Polyploidy or Whole Genome Duplication (WGD) is an ubiquitous and influential force in plant evolution, both on macro- and micro-evolutionary timescales. [1, 2] WGD could have played a crucial role in angiosperm evolution and in the diversification of their main lineages.

**Funding:** The author(s) received no specific funding for this work.

**Competing interests:** NO authors have competing interests.

Indeed, some ancient WGD events are associated with high rates of diversification, a remarkable example of which is related to the origin of *Asteraceae*, a family of angiosperms containing one of the largest number of species [3–6].

Polyploidy can arise in two ways. Allopolyploidy takes place when the genomes are derived from different species, individuals that can be produced by interspecific hybridization. By contrast, autopolyploidy is the result of intraspecific WGD events and it has long been considered to be less common, although it is true that autopolyploids are more difficult to detect. Moreover, if the rate of tetraploid formation were comparable to the mutation rate, many more autopolyploids would be expected in nature [7,8].

Differences in the distribution of polyploids and diploids have long been noted [9]. Polyploid taxa are frequently found in plant communities worldwide and they seem especially abundant in challenging environments, such as at high latitudes, at high elevations or in arctic areas [9–14]. Such distributions are consistent with observations regarding the physiological changes in polyploid plants, and when changes in body size follow genome duplication, it is common to find that "morphological differences *could result in different adaptations of cytotypes*" (emphasis ours) [15,16]. Other central processes in plant physiology may also be affected by genome duplication and there is evidence that polyploids display a higher tolerance to water deficit [17] or higher photosynthetic rates [18]. One recurrent observation in natural polyploid plants is that they usually show improved growth and better adaptation to harsh environments, consequently displaying evolutionary advantages [19]. These observations explain the higher frequency of polyploids proposed in extreme environments [20].It is known that the rate of unreduced gamete formation increases under conditions of stress in plants [21, 22], such as extreme cold, highly fluctuating temperatures, nutritional and water stress, amongst others [21]. In areas where such challenging conditions prevail, small populations would be expected and this circumstance could further enhance the establishment of polyploids [23]. Interestingly, polyploids often have a higher rate of self-fertilization, a higher capacity to propagate clonally and a perennial life-cycle. All these traits would improve their chances of establishing stable populations in areas where conditions are non-optimal [20,24,25]. The higher frequency of polyploid individuals in non-optimal areas or extreme bioclimatic conditions, if true, could be relevant to the origin of allopolyploid species. Moreover, it is known that hybridization between distinct autopolyploids directly produces allopolyploids [21] and according to this observation, the presence of a high frequency of compatible autopolyploids in an area could trigger the formation of stable populations of the corresponding allopolyploid hybrid.

Ecological Niche Models (ENMs) along with Species Distribution Models (SDMs) [26] are useful tools to visualize the ecological preferences of a species within a specific area and they have been used successfully to examine a wide range of ecological issues. In a number of studies they have also been used to evaluate the effects of climate change on ecosystems and communities, or the effectiveness of conservation plans [27–36]. In terms of species distribution models (SDMs), all the above observations could mean that the frequency of polyploid individuals increases at the periphery of species ranges [37–40], where the suitability for the species is low but not zero. Such Low Suitability Areas (LSAs) would harbor populations with a higher frequency of autopolyploids. If we consider two interbreeding species, we would expect a higher formation of allopolyploids in areas where the periphery of both ranges overlap. If there is a higher frequency of autopolyploids, the probability of producing non-reduced gametes would also be higher [22], increasing the probability of fertile interspecific hybrids arising. Thus, we could consider these areas as more favorable sites to establish stable allopolyploid populations (Fig 1), and such locations could act as potential formation centers or cradles for the hybrid species.

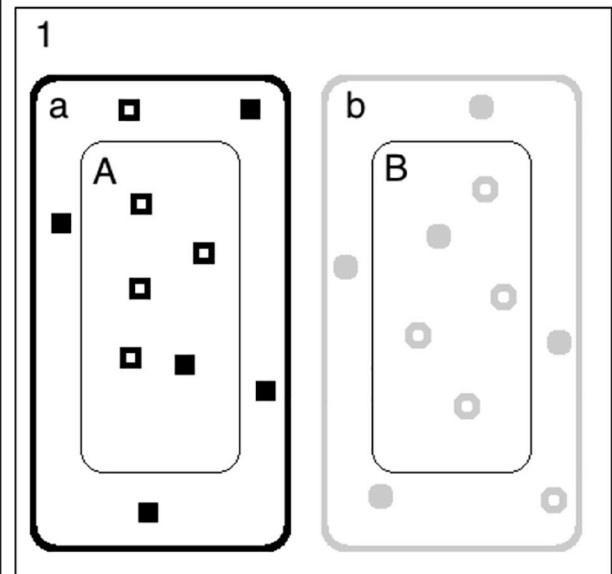
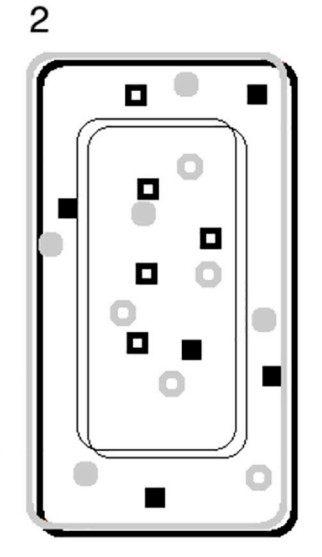
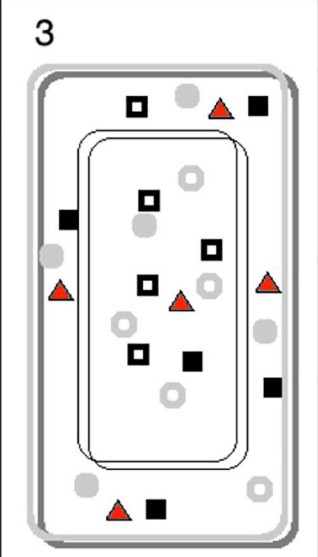

**Fig 1. A possible mechanism of cradle formation.** We show two closely related species with an allopatric distribution. Both of them have a high suitability area around the centroid of their ranges (regions *A* and *B* in 1) and a lower suitability area (LSA) around their ranges (areas *a* and *b* in 1). According to a number of observations (see text), a higher frequency of autopolyploids (solid squares and circles) could be expected in the peripheral regions of both ranges (the LSAs). The LSAs will overlap if both species have some degree of sympatry and hence, there would be regions with a stronger presence of autopolyploids emanating from both species. This circumstance would favor the formation of fertile tetraploid hybrids (red triangles, species C in 3). We considered such areas as potential cradles or potential formation centers for the hybrid species. Such hybrids could also arise at other locations where the parental taxa could interbreed, although a lower rate of hybrid formation would be expected. This is a possible scenario for the speciation of *G. bermejoi*, where the A and B species would be its parental *G. struthium* subsp. *struthium* and *G. tomentosa*.

*Gypsophila bermejoi* G.López is an allopolyploid species derived from the parental *G. struthium* L. subsp. *struthium* and *G. tomentosa* L [41]. All these plants are gypsophytes endemic to the Iberian Peninsula, and they are of particular ecological, evolutionary and biochemical interest. Some evidence points to a recent origin of the current populations of *G. bermejoi* [42,43]. Indeed, based on ENMs and SDMs it has been shown that the parental taxa of *G. bermejoi* had a high degree of sympatry during the Last Glacial Maximum (LGM), although the climatic conditions may have repressed the formation of this species because of its very narrow bioclimatic niche at that time [42,43]. In the present work gypsum soil locations included in areas where the LSAs of the parental species of *G. bermejoi* overlap were determined. According to our hypothesis such locations could be potential cradles for *G. bermejoi* speciation (Fig 1) Thus, we located a set possible centers where the climatic conditions during the (LGM) and the mid-Holocene (MH) period would favor the formation of this species, and then, we assessed the evolution of the bioclimatic conditions at those locations to determine which populations could have survived. Finally, we compared these results with the current

locations of *G. bermejoi*. Our main goal was to find out if this working hypothesis could to some extent predict the current distribution of the species.

## 2 Material and methods

### 2.1 Study area and species

All the species included in this work are endemic to the Iberian Peninsula, which is located in the southwestern part of Europe, between latitudes 36˚00'08"N—43˚47'38"N and longitudes 9˚29'00"O—3˚19'00"E. With an area of approximately 597,000 km$^2$, this is the second largest peninsula on the continent, and it includes most of the territory of Spain and Portugal. In this study, we focused on the gypsum steppes within the Peninsula, containing the gypsum out-crops in this territory and covering a total extension of 32,487 km$^2$ (5.4% of the Peninsula) [44]. These habitats are concentrated in the eastern half of the Peninsula, under the influence of a Mediterranean climate.

Our research focuses on the speciation of *Gypsophila bermejoi*, a natural hybrid derived from *G. struthium* subsp. *struthium* (2n = 34) and *G. tomentosa* (2n = 34) [41]. This genus is included in the *Caryophyllaceae* family of plants, and it is an allopolyploid species (2n = 68) found on verges and slopes in areas of gypsum soil [44]. While the range of *G. bermejoi* over-laps that of its parental taxa to some degree [45], the ecological niche of *G. bermejoi* shows noticeable differences [43]. Gypsum soils are harsh environments for angiosperms due to their extreme xericity, severe nutritional imbalances and the presence of a crusty surface [45,46]. In biochemical terms, gypsophytes can produce a wide range of secondary metabolites and they have a tendency to accumulate certain minerals. The production of saponins and antioxidants, like that of phenols and flavonoids, has been studied widely in *Gypsophila* [47–49]. As a result of the particular ecological conditions of the gypsisols, the peninsular gypsum outcrops harbor plants that are highly adapted to their environment, and endemic species are common at these sites. This circumstance and the patchy nature of these habitats makes these species a good model to study plant evolution and speciation processes [46].

### 2.2 Data sources

According to a number of works [50,51] the use of only one algorithm is not the ideal strategy to model ecological niches and potential species distributions. In this research we performed ensembles based on four algorithms: General Linear Models (GLM) [52], Random Forests (RF) [53–55], Mlutivariate Adaptative Regression Splines (MARS) [56,57] and Maximum Entropy (MaxEnt) [58–61]. All these algorithms require two types of data for the study area: species occurrence and environmental data. Here, we used species occurrence data from the Global Biodiversity Information Facility (GBIF) database [62] and a set of WorldClim bioclimatic variables. WorldClim is a database with sets of global climatic data that can be used for ecological modeling and mapping (gridded climate data), with a spatial resolution of about 1 km$^2$. The internet portal (www.worldclim.org) offers publically data for 19 bioclimatic variables all them derived from temperature and precipitation data from meteorological stations. [63].

An important aspect of ENM and SDM studies is the uncertainty carried in the data, especially when extrapolating models for future or past conditions, since this data is a result from Global Circulation Models (GCMs). There is considerable variation for some of the bioclimatic variables among the different GCMs. Thus, we used bioclimatic maps for the MH period and LGM generated through three different Global Circulation Model: the Community Climate System Model (CCSM4), the Model for interdisciplinary Research of Climate (MIRO-C-ESM) and the Max Planck Institute for Meteorology Earth System Model (MPI-ESM) [64–

66]. In projections to the LGM, this is particularly important for species highly sensitive to precipitation variables in temperate areas [67, 68]

Modeling species distributions requires correlations between the predictors to be minimized [61]. From a set of available predictors, we selected those with the lowest Pearson correlation coefficient using a clustering algorithm to produce a dendrogram that groups the variables according to their degree of correlation. A jackknife test of the contribution of the variables to the model was also performed. After combining the information obtained through correlation dendrogram and jackknife for variable contribution, and taking into account the abiotic factors determining the vegetation patterns in a semi-arid Mediterranean landscape [69,70], we selected from the less correlated variables those with the highest contribution, and one related with precipitation values during the dry season (Table 1).

## 2.3 Modeling species distributions

We have used the SDM R package [71,72] to model the potential distribution maps of the studied species. It provides a reproducible and extensible platform for species distribution modeling, suitable for reproducible research. This package includes tools enabling ensembles of models to be performed, model evaluation using different approaches, and SDM projections in space and time [72]. In our case, we used a set of algorithms that included General Linear Models (GLM), Random Forests (RF), Mlutivariate Adaptative Regression Splines (MARS) and Maximum Entropy (MaxEnt), all them currently implemented in R language [71]. For an explication of the methods see [52,61]. These tools essentially assess the probability of the presence of a species in function of the environmental variables selected. The inputs are data files on the presence of the species and a series of maps of the environmental variables considered to be significant. The results can be interpreted as an index of suitability for the particular species to that habitat [59].

The models were fitted and used to generate multiple replicates (30) of each method through partitioning using the bootstrap method. Weighted average ensambles of these models were performed for each species. Then, we obtained species distribution projections for the LGM and the Mid Holocene climatic conditions using three different General Circulation Models: CCSMS, MIROC-ESM and MPI-ESM. Again, an ensemble was performed to get the consensus forecast map.

We dealt with autocorrelation issues by eliminating any redundant presence in each pixel on the scale of the bioclimatic variables used. The original numbers of occurrence records and the numbers left after this step (at a resolution of 2.5 minutes) were: 138 and 60 respectively for *G. bermejoi*, 256 and 120 for *G. struthium* subsp. *struthium* and 349 and 99 for *G. tomentosa*. The uncertainties were assessed and mapped by studying the Standard Deviation of the ensembles (see supplementary materials 1 and 2). The study focused on the effect of climatic oscillations and as such, we used only bioclimatic variables to generate the models. Given the strict gypsophyllic nature of these plants [44], they are only present on gypsum soils and their suitability on other soils is "0". Therefore, we applied the models with climate variables to the

**Table 1. List of the environmental variables selected to implement both the MaxEnt species distribution models.**

| Variable | Source | Resolution |
|---|---|---|
| **bio4**—the dispersion of seasonal temperatures | WorldClim | 2.5' |
| **bio6**—the minimum temperature in the coldest month | WorldClim | 2.5' |
| **bio13**—the rainfall in the wettest month | WorldClim | 2.5' |
| **bio14**—the rainfall in the driest month | WorldClim | 2.5' |

entire Iberian Peninsula and we then subtracted the gypsum soil territories to evaluate the suitability areas. We validated the models by means of the Area Under the Curve (AUC), and the-True Skill Statistic (TSS), [73–75].

## 2.4 An approach to find potential *G. bermejoi* cradles

As pointed out previously, a number of observations seem to support the hypothesis of a higher frequency of polyploid individuals in the peripheral domains of the ranges for a given species [37–40], where the suitability for the species is weaker. If this statement is true, higher autopolyploid frequencies would be expected at the periphery of the potential areas predicted by the models. Here we used this idea to predict possible centers of *G. bermejoi* formation. As indicated, our hypothesis is that the gypsum outcrops where the LSAs of both parental taxa overlap (suitable soils with higher frequencies of *G. struthium* subsp. *struthium* and *G. tomentosa* autopolyploids) would be potential cradles for the allopolyploid species *G. bermejoi*. As such, we assessed whether the results obtained were consistent with the current distribution of *G. bermejoi*.

Our approach to find potential cradles is summarized as:

**a Generating species distribution models (SDM).** The ensemble modeling approach was used to produce suitability forecasts based on a series of bioclimatic variables carefully selected (see above) and four different algorithms (GLM, RF, MARS and MaxEnt). These models were then projected onto the geographical space of the Iberian Peninsula in the current climatic conditions, and in two past climatic ages: the LGM and MH period. For the LGM and MH we again performed an ensemble based on three General Circulation Models (CCSMS, MIRO-C-ESM and MPI-ESM). We generated these models for *G. bermejoi* and for its parental taxa, *G. struthium* subsp. *struthium* and *G. tomentosa*.

**b Obtaining a statistical distribution of suitability values.** To achieve our goal it is crucial to define what is "low" suitability. To do so, we carried out an exploratory analysis of the suitability values predicted by the models under the current climatic conditions for both parental taxa. A statistical distribution for these bioclimatic suitability values was obtained, considering only the sites where they are currently present. In other words, we took the readings of the suitability values predicted by the models at the locations where the species is currently found. Our goal here was to define the lowest suitability values predicted by the model at the sites where both taxa occur. (*G. struthium* subsp. *struthium* and *G. tomentosa*). It is important to bear in mind that by doing so, we obtain the lowest suitability values predicted by the SDM for the parental taxa at locations where those plants really grow (Fig 2A).

**c Defining Low suitability intervals.** At this point we need to establish a threshold to define what can be considered "low". To do so, we used the range between the minimum and the first quartile of the statistical distributions for the suitability values of the parental taxa (Table 2). We considered a low suitability interval to be that defined between the minimum and the first quartile of the suitability values predicted by the SDM at the current locations of both taxa (*G. tomentosa* minimum = 0.07603, first quartile = 0.52207 and *struthium struthium* minimum = 0.1306, first quartile = 0.4447). It is important to bear in mind that our aim is to assess the suitability interval that could enhance the formation of allopolyploids. In other words, the suitability values favoring a higher frequency of autopolyploids for at least one of the parental taxa. We used the same interval for both taxa and we established the interval between the lowest minimum and the highest first quartile (Fig 2A). We have to consider this step by step, whereby below the lowest minimum no taxon is present and obviously, we must discard that interval. Between both minima, we had low suitability for one taxon and no presence for the other, making interbreeding impossible. The interval defined by the *G. struthium*

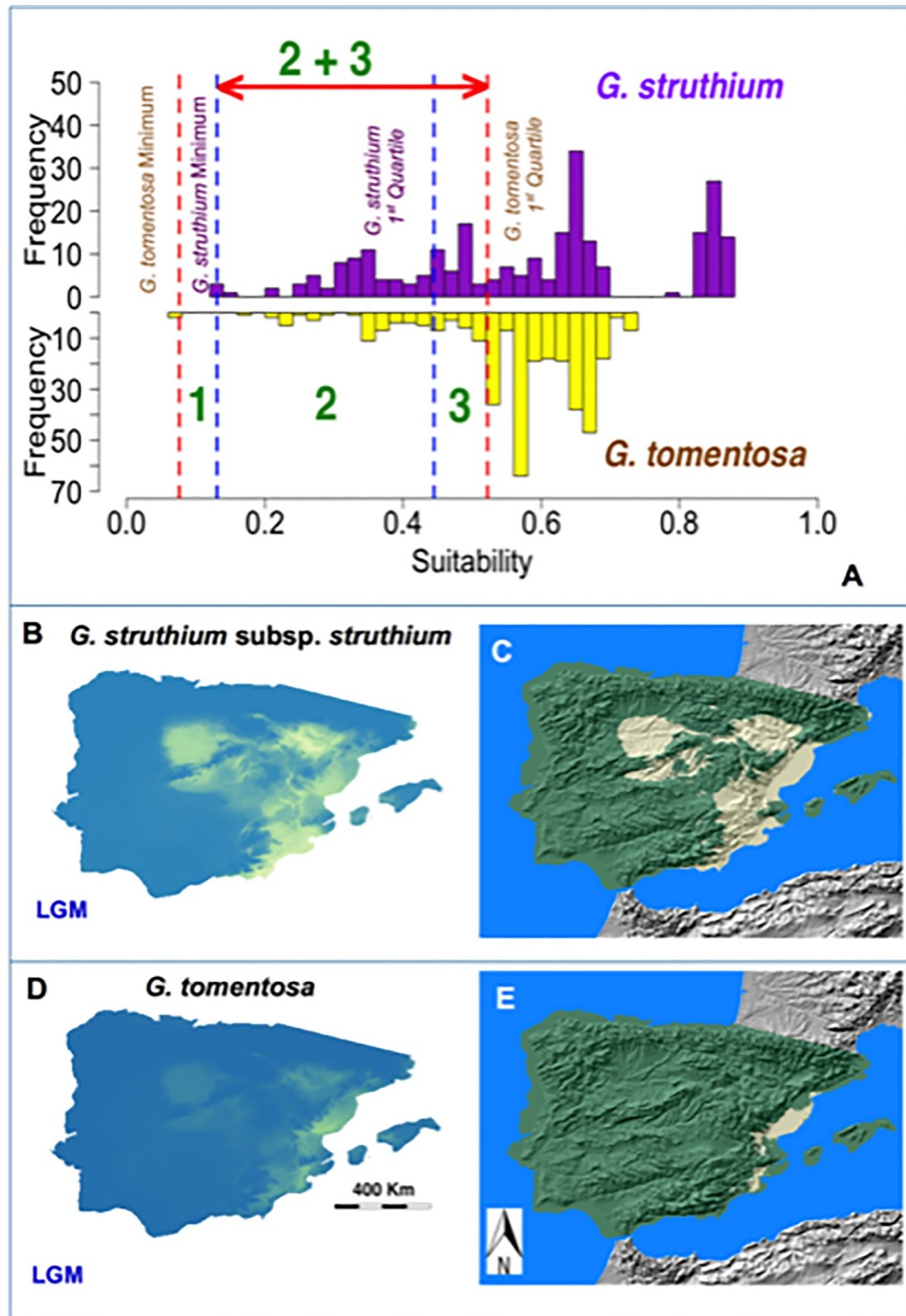

**Fig 2. Low Suitablility Areas (LSA) for *G. struthium* subsp. *struthium* and *G. tomentosa*.** To establish the intervals of low suitability, we need to know the suitability values predicted by the models at the locations where these taxa are present under current climatic conditions. For each taxon, we considered low suitability values to be all those between the minimum and the first

quartile. We tried to assess the suitability values in which a high frequency of autopolyploids is expected for both species. In Fig 2A, the suitability value histograms for both taxa are shown, with the minimum and first quartiles indicated by colored lines. In interval 1 (the minimum between *G. tomentosa* and *G. struthium* subsp. *struthium*), there is low suitability for *G. tomentosa* but no presence of *G. struthium* subsp. *struthium* (making interbreeding impossible). In interval 2 (between the minimum and the first quartile of G. struthium subsp. struthium) there is low suitability for both taxa and according to our hypothesis, a higher frequency of autoplyploids might be expected. Interval number 3 has low suitability for *G. tomentosa* but not for *G. struthium* subsp. *struthium*. If we accept the situations for low suitability, there could be a higher frequency of *G. tomentosa* autopolyploids in this interval but not of *G. struthium* subsp. *struthium*. In these conditions a higher frequency of triploid individuals would be expected, which could also enhance the formation of the *G. bermejoi* tetraploid hybrid. As a consequence, we considered the 2+3 interval to be favorable for the formation of *G. bermejoi*, between the higher minimum (that of *G. struthium* subsp. *struthium*) and the higher first quartile (that of *G. tomentosa*). Fig 2B and 2D show the MaxEnt models during the LGM for *G. struthium* subsp. *struthium* and *G. tomentosa*, respectively, whilst Fig 2C and 2E show the LSAs for both taxa (suitability values between 0.07603 and 0.52207).

subsp. *struthium* minimum and its first quartile represents a low suitability for both parental taxa (Fig 2A). In these conditions higher frequencies of autopolyploids could be expected for both taxa, as well as a higher rate of allopolyploid formation. However, we also considered the interval between the first quartiles of *G.struthium* subsp. *struthium* (0.4447) and *G. tomentosa* (0.52207). If the working hypothesis were correct, there would be a higher frequency of auto-polyploids in this interval for just one taxon. However, according to the triploid bridge mechanism [76,77], this interval could also facilitate the formation of allopolyploids. As a consequence, we consider that the bioclimatic suitability interval between the minimum of *G. struthium* subsp. *struthium* (0.1306) and the first quartile of *G. tomentosa* (0.52207) could favor the formation of *G. bermejoi* allopolyploid species. Such intervals correspond to a series of pixels in the maps defining the LSAs for each taxon. The next logical step is to locate those regions in the geographical space and to find where the LSAs for the parental taxa overlap.

**d Low Suitability Areas in the geographic space for the LGM.** Once the low suitability interval is defined, a GIS can be used to establish the LSAs for the parental species of *G. bermejoi* during the LGM, for which we used QGIS [78]. From the potential area predicted for *G. struthium* subsp. *struthium* and *G. tomentosa* during the LGM (Fig 2B and 2D), we selected those areas with suitability values between 0.1306 and 0.52207 (Fig 2C and 2E). According to our working hypothesis, those areas should have more autopolyploid plants and if so, a higher frequency of allopolyploids would be expected in the locations where both these areas overlap (Fig 3B and 3D). A stronger presence of non-reduced gametes in such populations would facilitate the formation of the fertile hybrid *G. bermejoi*. Thus, we used QGIS to establish where the LSAs for the parental taxa overlap.

**Table 2. Models evaluation using AUC and TSS.** AUC values next to one and TSS values above 0.6 are considered good and 0.2–0.6 to be fair to moderate.

| | MaxEnt | RF | GLM | MARS |
|---|---|---|---|---|
| ***G. bermejoi* AUC** | 0.97 | 0.99 | 0.94 | 0.97 |
| ***G. bermejoi* TSS** | 0.87 | 0.93 | 0.86 | 0.86 |
| ***G. tomentosa* AUC** | 0.95 | 0.98 | 0.87 | 0.95 |
| ***G. tomentosa* TSS** | 0.81 | 0.89 | 0.7 | 0.81 |
| ***G. struthium* AUC** | 0.95 | 0.98 | 0.89 | 0.95 |
| ***G. struthium* TSS** | 0.78 | 0.88 | 0.7 | 0.77 |

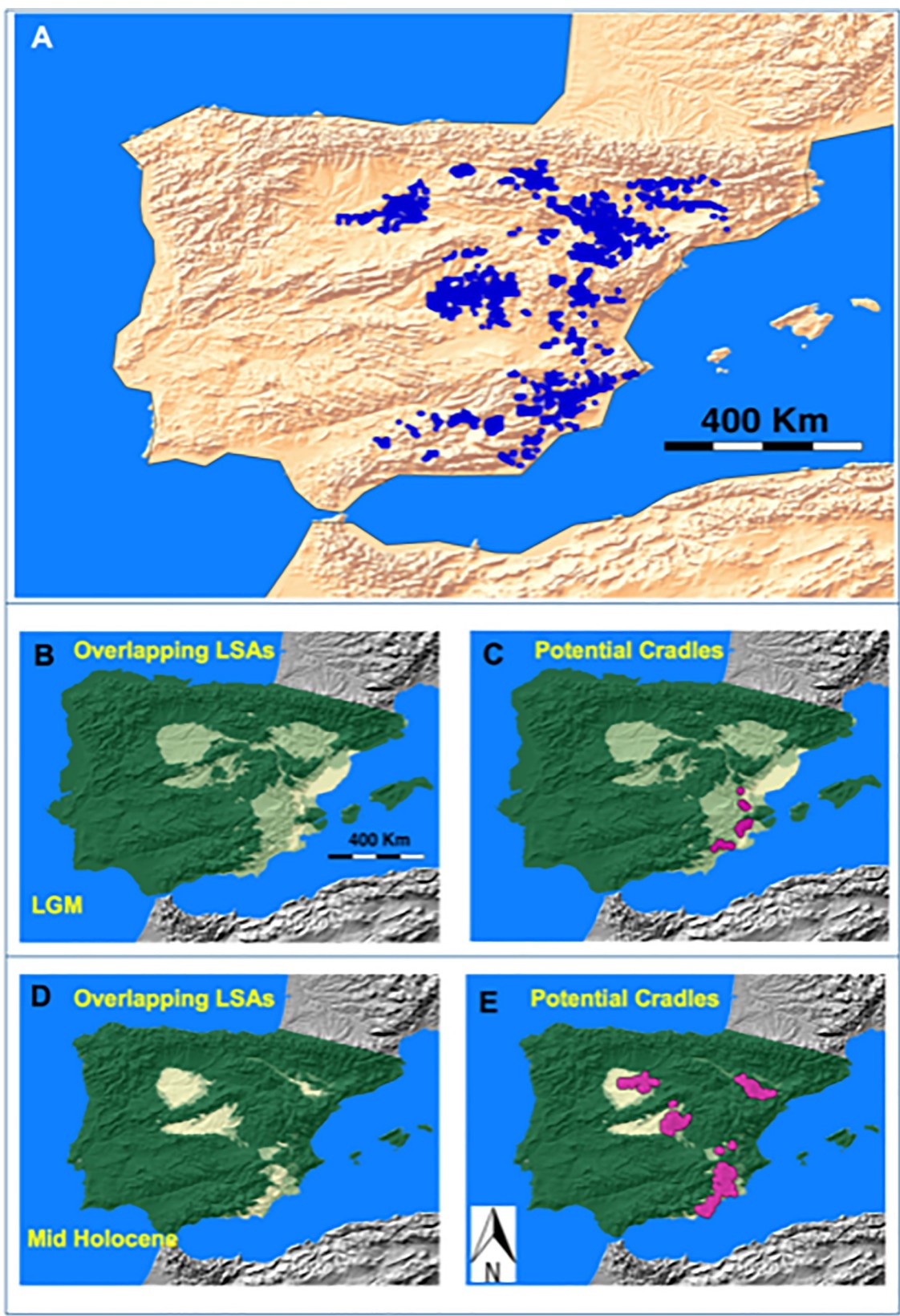

**Fig 3. A digital version of the gypsum soils maps published by Mota *et al.*** (2011) was used to plot the sites of gypsum soils on the map *(3A)*. During the LGM, the LSAs of the *G. bermejoi* parental taxa overlapped at some locations due to the high degree of sympatry of both taxa (yellow colored areas in *3B* and *3C*). According to our hypothesis, higher frequencies of autopolyploids and non-reduced gametes would be expected in those areas. The three taxa studied are strict gypsophytes and hence, we filtered all the gypsum outcrops included in the areas where the LSAs for both parental taxa (*G. struthium* subsp. *struthium* and *G. tomentosa*) overlap *(3C)*. At those sites, there would be a higher probability of establishing thriving populations of the fertile *G. bermejoi* hybrid and thus, we consider such places as potential cradles for this species. *3D* shows The overlapping LSAs of the parental *G. bermejoi* taxa during the MH period and *(3E)* the potential cradles for the same climatic period.

**e Filtering for locations with suitable soils: Locating the LGM cradles.** All the species considered in this work are strict gypsophytes. So far we have located the areas where bioclimatic suitability could favor the formation of *G. bermejoi* but we have yet to take into account the special edaphic conditions required by this plant. To select the sites with suitable soils, we used a digital version of the gypsum soils maps published by Mota *et al.* (2011) [44]. Using an algorithm implemented in R programming language [71] and QGIS [78], we filtered the sites with gypsum outcrops from the areas where the low bioclimatic suitability of the parental taxa overlap (Fig 3B and 3C). We consider these locations as potential cradles for *G. bermejoi* and in such places, the presence of two concomitant factors could favor the establishment of stable populations of this species: suitable soils and according to the working hypothesis, a high frequency of polyploid individuals from the parental *G. bermejoi* taxa.

**f Locating potential cradles for the Mid-Holocene period.** The SDM models show dramatic changes in the potential distribution for the species studied during the MH period. New potential bioclimatic areas mean new LSAs and hence, new potential cradles. To locate these potential cradles, we repeated the protocol using the models for the MH period (Fig 3D and 3E): the LSAs for the MH period can be seen in the supplementary materials section (S3).

## 2.5 Predicting the expected distribution of *G. bermejoi* under current climatic conditions

Having determined the potential *G. bermejoi* cradles, it is possible to forecast its expected current distribution by studying the evolution of the bioclimatic suitability of this taxon at such locations or, in other words, by foreseeing the fate of such potential cradles. Our goal here is to find out which of these locations could have maintained stable populations of *G. bermejoi* despite the dramatic climatic changes that have occurred in the past 25,000 years. By doing so, we could assess which potential cradles might have survived such climatic oscillations and could currently harbor *G. bermejoi* populations. Hence, we assessed in which of these cradles bioclimatic suitability remained reasonably high for *G. bermejoi* as follows:

**a Defining a survival threshold for *G. bermejoi*.** A "survival" threshold was defined in order to establish which of the possible cradles could have maintained *G. bermejoi* populations up to the present day. To do so, we generated a model for *G. bermejoi* under current climatic conditions and we took the suitability values at the points of occurrence of this species. The resulting distribution is shown as a histogram in Fig 4A to 4D (lower histograms) and the minimum suitability value of this distribution was used as a "survival" threshold. We also evaluated the effects of different thresholds in the results (see discussion).

**b Evolution of the bioclimatic suitability conditions for *G. bermejoi* at the cradles.** The evolution of bioclimatic suitability of the potential cradles as a whole was studied from the LGM to the present (Fig 4). We treated the potential cradles generated during the LGM (Fig 4A and 4B) separately from those generated in the Mid-Holocene (Fig 4C and 4D). The former were studied at three moments: LGM, MH period and current climatic conditions. Obviously, we only studied suitability during the MH and the current climate for the MH cradles. We

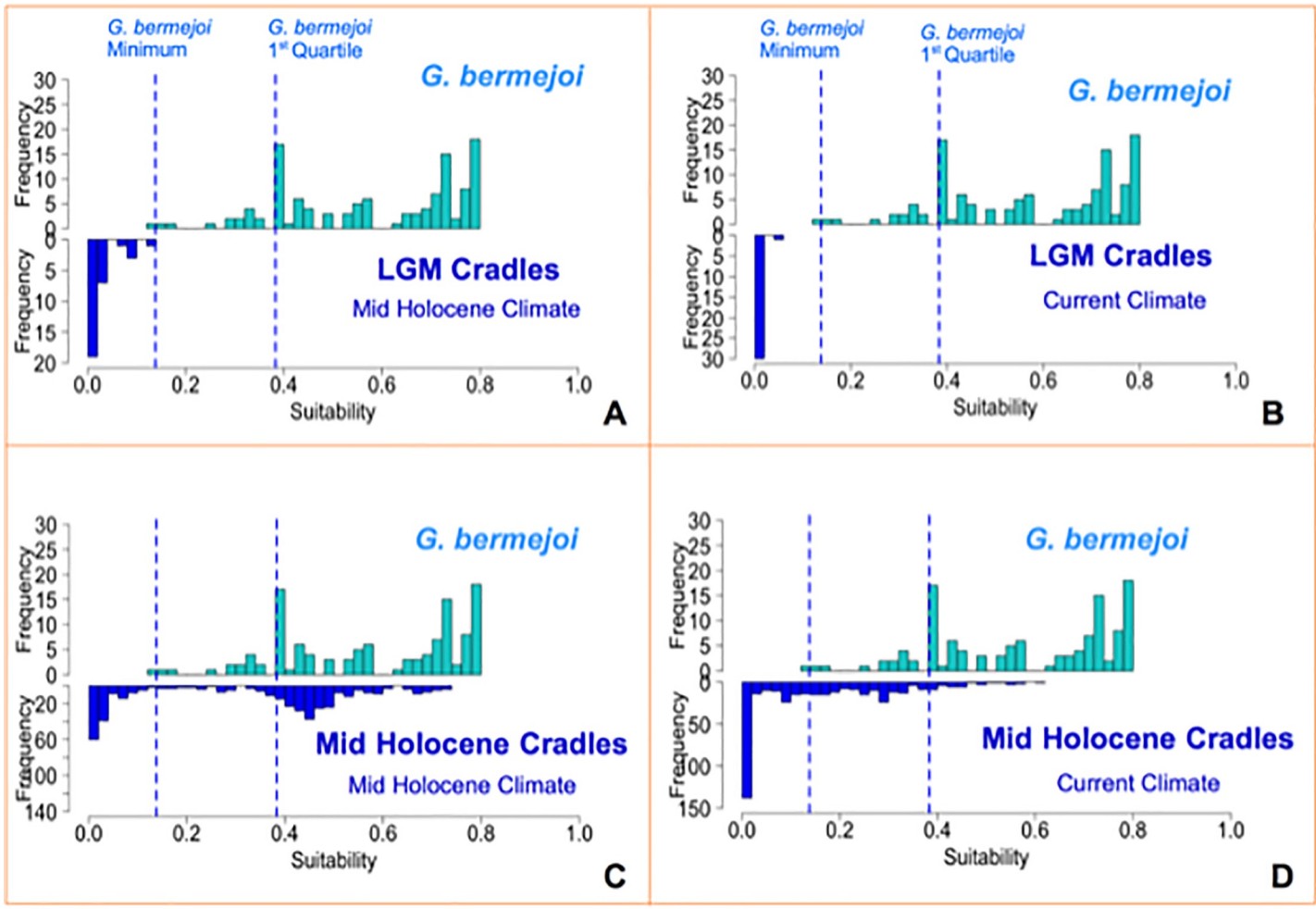

**Fig 4. The bioclimatic suitability of *G. bermejoi* cradle locations in the LGM and Mid-Holocene periods were evaluated in the Mid-Holocene and current climate conditions.** The resulting statistical distributions were plotted as the dark blue histograms, while the pale blue histogram corresponds to the suitability values of the model at the sites the plant occurs under current climate conditions. The minimum value of this distribution was used as a survival threshold for this species. Comparing both histograms, we can see how many potential cradles maintain a bioclimatic suitability below this minimum and thus, *G. bermejoi* could not endure the climatic conditions at these sites. *(4A)* shows the comparison between *G. bermejoi* suitability values and those of the LGM cradles under Mid-Holocene climatic conditions and *(4B)* those for the current climatic conditions. *(4C)* compares the *G. bermejoi* suitablity values with those of the Mid-Holocene cradles during the Mid-Holocene and *(4D)* under current climatic conditions. In LGM cases *(4A* and *4B)*, the vast majority of cradles produced very low suitability values for *G. bermejoi* and it is difficult to consider the survival of this species at such locations. In contrast, for Mid-Holecene cradles it seems to be a number of surviving populations during that period *(4C)* and some of them to the present *(4D)*.

then took the suitability values of *G. bermejoi* produced by the models at the LGM cradles for the MH and current climatic conditions.

Comparative histograms were generated to get an idea of the potential cradles where the suitability values for *G. bermejoi* remained below the minimum for different climatic periods (Fig 4), sites at which *G. bermejoi* populations would be doomed. This approach also enabled us to filter the surviving cradles at different moments in the past and under the present climatic conditions.

**c Filtering the potential cradles in the geographical space: an expected current distribution prediction.** Two series of maps were generated using QGIS [78] to depict the fate of both the LGM and MH potential cradles (Fig 5A to 5D). A color code reflecting the bioclimatic suitability values at those specific locations was used to show how favorable such sites were for

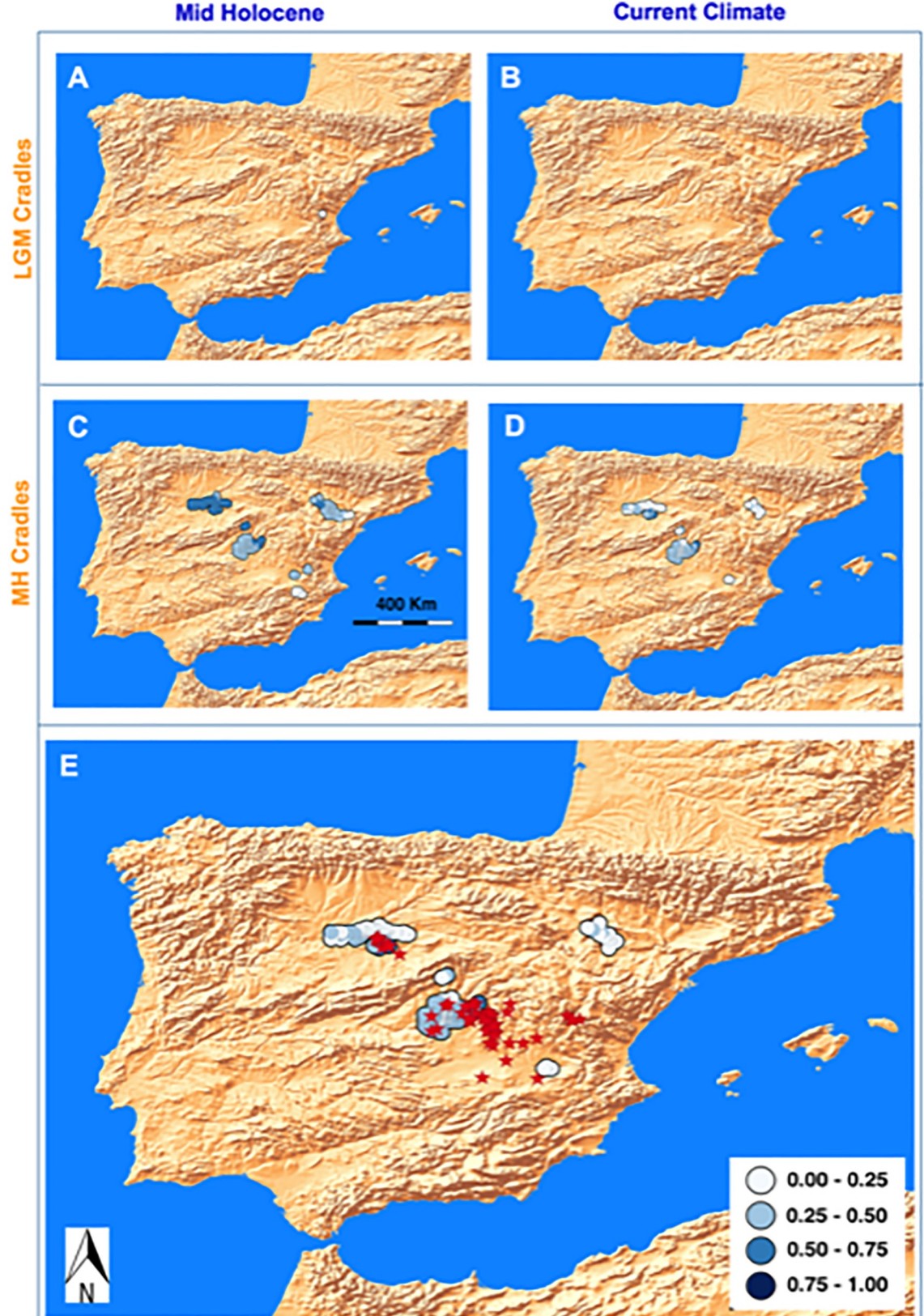

**Fig 5. Fate of the potential *G. bermejoi* cradles and expected current distribution of this plant according to our hypothesis.** The quaternary climatic oscillations over the past 25,000 years have produced dramatic changes in bioclimatic suitability for this species. (5A) Shows the LGM cradles where *G. bermejoi* populations are expected to survive during the MH and current (5B) climatic conditions. We repeated this analysis for the potential cradles produced during the MH period and the results are shown in (5C) and (5D). This approach enabled us to produce a predicted current distribution for *G. bermejoi*, assuming no dispersal from the original cradles. This expected current distribution is shown in (5E), along with the actual presences of *G. bermejoi* (red triangles).

*G. bermejoi*. All the cradles with suitability values below the minimum were removed because *G. bermejoi* populations could not endure such bioclimatic conditions. Our goal here was to assess which cradles could have harbored *G. bermejoi* populations up to the present. It is important to note that on this occasion we used the suitability values for *G. bermejoi* whilst we used those of its parental taxa to locate the potential cradles.

Due to the dramatic climatic changes in the Iberian Peninsula over the last 25,000 years it is possible that the bioclimatic suitability for *G. bermejoi* may drop abruptly in some potential cradles, areas in which the *G. bermejoi* populations will be doomed. Using these maps it is possible to find out in which potential cradles the conditions might maintain thriving *G. bermejoi* populations to date (Fig 5B and 5D). We applied the process for both LGM and MH cradles and in doing so, we can predict the current distribution of *G. bermejoi* based on an assumption of no dispersal, a prediction that can be compared with the actual distribution of this species (Fig 5D).

## 3 Results

### 3.1 Models validation

The AUC and TSS values obtained for the different algorithms and taxa are summarized on Table 2. One must bear in mind that AUC values next to one and TSS values above 0.6 considered good and 0.2–0.6 to be fair to moderate.

### 3.2 Location and fate of potential LGM and Mid-Holocene cradles for G. *bermejoi*

We had adopted a two stage strategy to predict the current distribution of G. *bermejoi* based on our hypothesis. First, we located the potential cradles according to the distributions of both parental taxa (*G. struthium* subsp.*struthium* and *G. tomentosa*) during the LGM and the MH periods, and we then assessed in which of those possible cradles the climatic conditions remained reasonably well suited to *G. bermejoi* survival. We obtained several possible centers of formation during the LGM, located in the eastern region of the Iberian Peninsula,(Fig 3C). However, the bioclimatic suitability for *G. bermejoi* is almost 0 at all of these sites during the MH and only one population seems to survive that climatic period (Fig 5A) but no one to the current climatic conditions (Fig 5B).

The dramatic change in the potential distribution provoked by the climatic transition to the MH period favored different potential cradles for *G. bermejoi*. In this case, the potential cradles were more or less clustered at five locations: the Central, Eastern and South Eastern Region of the Iberian Peninsula; the Ebro Valley; and interestingly, a site in the North West of the peninsula, beyond the Central Mountain System that represents an important geographical barrier. Most South Eastern areas were ill-fated in terms of *G. bermejoi* survival and hence, those below the survival threshold were not considered further (Fig 5C). During the MH period, we saw high bioclimatic suitability at several locations in the Ebro Valley and in the Central, Eastern and North western regions. After this period of benign climatic conditions for *G. bermejoi* at many sites (Fig 5C), its suitability decreased at many of these locations under current

climatic conditions. At present, only some of the cradles in the central and north western sites maintain suitability values that would be compatible with the survival of *G. bermejoi* populations (Fig 5C).

The surviving cradles can be considered (Fig 5D), along with the sites with gypsophyte suitable soils (Fig 3A). Most of these gypsum outcrops remain unoccupied by *G. bermejoi* under current climatic conditions, although the models show that this was probably not the case in the past (Fig 3C and 3E). The surviving cradles can be interpreted as the expected distribution of *G. bermejoi* under conditions where there is absolutely no dispersion (Fig 5E). The occurrence of *G. bermejoi* is shown along with this expected distribution in Fig 5E for comparison.

## 4 Discussion

The results presented here suggest that the current distribution of *G. bermejoi* could be explained, at least in some degree, by our hypothesis. Accordingly, we outline some potential key events concerning the recent evolutionary history of this taxon. In general terms, the bioclimatic suitability for *G. bermejoi* evolved considerably over the last 25,000 years. From the almost null bioclimatic suitability in the LGM, there was an improvement during the MH period that subsequently decreased as the current climatic conditions were established. Our results show how this trend could have affected the possible formation of *G. bermejoi*, which would not have been present during the LGM. By contrast, several thriving populations of this species could have been established during the benign MH period, thereafter declining as the current climatic conditions set in established.

If we consider the LSAs of the parental taxa during the LGM, we see that they overlap at a number of sites with suitable soils. According to our hypothesis, those areas would harbor higher frequencies of autopolyploids of both taxa and they would therefore be potential cradles for this species. While we cannot rule out the formation of *G. bermejoi* in places other than these, the formation rate of hybrid individuals would be lower, as would the probability that stable populations could thrive there. It seems reasonable to accept that the climatic transition to the MH period triggered the formation of the first *G. bermejoi* populations at such sites (Fig 3C). We believe that hybrid populations would have blossomed at such locations as soon as the LGM climate changed.

During the MH period, new climatic conditions generated other potential areas for the parental taxa by producing different LSAs for *G. struthium* subsp. *struthium* and *G. tomentosa* and hence, new potential cradles for *G. bermejoi*. We think that the further climatic changes filtered those formation centers and our results indicate that only some of them maintained the bioclimatic conditions for *G. bermejoi* populations to survive. Such surviving populations are located in the Ebro Valley, the central, eastern and south eastern sectors, and interestingly, in the North West beyond the Central mountain range. Most of the observed sites of *G. bermejoi* are not too distant from the former ones and we think that this distribution is consistent with our hypothesis.

The presence of *G. bermejoi* populations in North west of Spain, beyond the Central System seems difficult to explain in terms of migration. This mountain range is a geographic barrier extending in ENE-WSW direction, with summits reaching heights of 2500 m [79] and constituting the boundary between two large tertiary basins: the Duero Basin to the North and the Tagus Basin to the South. This mountain range was the outcome of deformation, and it developed through triaxial compression that occurred during the Oligocene and the Lower Miocene period (*circa* 25 million years ago) [79–80]. Our hypothesis points to an *in situ* origin of these populations, because this area matches one of the MH cradles that maintained some degree of

climatic suitability up to the present. Thus, the presence of *G. bermejoi* there could be understood in the framework of our hypothesis.

Our predictions include some locations in the Ebro valley, although we feel that their low suitability values (bioclimatic suitability < 0.25) might rule out these sites as potential cradles. We used also other values of climatic suitability as survival thresholds. The first quartile and its mean with the minimum were also considered. These values decrease markedly the number of surviving populations both in the MH and the current climatic conditions. The histograms showing the bioclimatic suitability at the sites of *G. bermejoi* occurrence according to the models suggest an extremely low frequency of plants below 0.25. Indeed, the species is not currently found in the central Ebro Valley. Moreover, *G. bermejoi* is currently quite dispersed across the Southern Plateau, a flat and windy region. Significantly, the anatomy and structure of *Gypsophila* seeds is appropriate for mid-range dispersal, far from the potential cradles, and these seeds can be released from their capsules by strong winds and animals [81]. Indeed, some invasive species of this genus (*G. paniculata*) apparently use a dispersal mechanism based on tumbleweed [82, 83].

To conclude, we believe that in conjunction with wind dispersal or other mechanisms, our hypothesis could to some extent explain the current distribution of *G. bermejoi*. However, many questions still remain unanswered regarding the recent evolutionary history of *G. bermejoi*. For instance, the connection between the gypsum outcrops and the dispersion capacity of *G. bermejoi* is unclear, and these data will be crucial to disentangle the recent evolutionary history of this taxon. However, our approach seems to shed some light on this issue and we encourage research into the genetic structure of the *G. bermejoi* populations, as well as cytogenetic studies to assess the rate and frequency of autopolyploid formation in current LSAs of *G. struthium* subsp. *struthium* and *G. tomentosa*.

## Conclusions

According to the results obtained here, we conclude that:

a.  The existence of populations of plants similar to those of the current *G. bermejoi* during the Last Interglacial period cannot be ruled out, although this question is beyond the scope of our study.

b.  The overlap between the LSAs for the parental *G. bermejoi* taxa during the LGM and the MH may have favored the formation of stable populations of this fertile hybrid soon after the end of this period. The gypsum outcrops in these areas could represent potential cradles or centers of formation for *G. bermejoi*.

c.  Only some of these potential cradles have retained suitable climatic conditions for this plant to date. These cradles that have survived to date may predict the expected distribution of *G. bermejoi*, assuming no dispersion. Such predictions seem to match the true distribution of *G. bermejoi*, particularly if we consider a degree of dispersion by wind or other means. This could explain the presence of this species in the central region of the Iberian Peninsula.

d.  The MH potential cradles could explain the presence of *G. bermejoi* populations to the North West of the Central mountain range. The presence of these populations is difficult to explain be means of migration, even though it is predicted by our model.

e.  The hypothesis of higher rates of autopolyploid formation in LSAs could to some extent explain the current distribution of *G. bermejoi*. Although further research will be required to clarify this issue, cytogenetic studies could address this hypothesis, for example assessing the autopolyploid frequencies of *G. struthium* subsp. *struthium* and *G. tomentosa* at their current LSAs.

## Supporting information

**S1 Fig. Uncertainty maps for the ensembles of the models generated by the four different algorithms (GLM, RF, MaxEnt and MARS) of *G. bermejoi*, *G. tomentosa* and *G. struthium* subsp. *struthium*.**
(TIFF)

**S2 Fig. The same for the ensembles performed from models using different climatic models for the projections to the MH and the LGM.**
(TIFF)

**S3 Fig. Low Suitability Areas (LSAs) for the parental taxa of *G. bermejoi* during the Mid Holocene.** For this climatic period, the MaxEnt models for *G. struthium* subsp. *struthium* and *G. tomentosa* are shown in *A* and *C* respectively. current distribution of this plant according to our hypothesis. *B* and *D* show the LSAs for both taxa during the Mid Holocene.
(TIFF)

## Author Contributions

**Conceptualization:** Miguel de Luis, Julio Álvarez-Jiménez, Carmen Bartolomé.

**Data curation:** Miguel de Luis.

**Formal analysis:** Miguel de Luis.

**Funding acquisition:** Julio Álvarez-Jiménez, Francisco Javier Rejos, Carmen Bartolomé.

**Investigation:** Miguel de Luis, Julio Álvarez-Jiménez, Carmen Bartolomé.

**Methodology:** Miguel de Luis, Julio Álvarez-Jiménez.

**Resources:** Carmen Bartolomé.

**Supervision:** Julio Álvarez-Jiménez, Francisco Javier Rejos, Carmen Bartolomé.

**Visualization:** Miguel de Luis.

**Writing – original draft:** Miguel de Luis.

**Writing – review & editing:** Miguel de Luis, Julio Álvarez-Jiménez, Francisco Javier Rejos, Carmen Bartolomé.

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
