## [Decision Letter · Decision Letter 0]

10 Sep 2019

PONE-D-19-14999

Using Species Distribution Models to locate the potential cradles of the allopolyploid Gypsophila bermejoi G. López (Caryophyllaceae)

PLOS ONE

Dear Mrs. Bartolomé,

Thank you for submitting your manuscript to PLOS ONE. After careful consideration, we feel that it has merit but does not fully meet PLOS ONE’s publication criteria as it currently stands. Therefore, we invite you to submit a revised version of the manuscript that addresses the points raised during the review process.

We would appreciate receiving your revised manuscript by 9th/Nov/2019. To enhance the reproducibility of your results, we recommend that if applicable you deposit your laboratory protocols in protocols.io, where a protocol can be assigned its own identifier (DOI) such that it can be cited independently in the future. For instructions see: http://journals.plos.org/plosone/s/submission-guidelines#loc-laboratory-protocols

We look forward to receiving your revised manuscript.

Kind regards,

Daniel de Paiva Silva, Ph.D.

Academic Editor

PLOS ONE

Additional Editor Comments:

Dear Dr. Bartolomé,

after a careful evaluation by two independent reviewers, one positive (major review) and one negative (rejection) that also raised interesting improvements that need to be taken care of, I believe you manuscript deserves a major review. Please focus on answering/changing the analyses suggested by both reviewers in the next review round.

Best regards,

Daniel de Paiva Silva

"NO"

Please provide an amended Funding Statement that declares *all* the funding or sources of support received during this specific study (whether external or internal to your organization) as detailed online in our guide for authors at http://journals.plos.org/plosone/s/submit-now

Please state what role the funders took in the study.  If any authors received a salary from any of your funders, please state which authors and which funder. If the funders had no role, please state: "The funders had no role in study design, data collection and analysis, decision to publish, or preparation of the manuscript."

3. Please include a copy of Table 2which you refer to in your text.

<h3>** **</h3>

4. We note you have included a table to which you do not refer in the text of your manuscript. Please ensure that you refer to Table 1 in your text; if accepted, production will need this reference to link the reader to the Table.

5. Please include captions for your Supporting Information files at the end of your manuscript, and update any in-text citations to match accordingly. Please see our Supporting Information guidelines for more information: http://journals.plos.org/plosone/s/supporting-information

Reviewers' comments:

Reviewer's Responses to Questions

**Comments to the Author**

1. Is the manuscript technically sound, and do the data support the conclusions?

Reviewer #1: Yes

Reviewer #2: Partly

2. Has the statistical analysis been performed appropriately and rigorously? 

Reviewer #1: Yes

Reviewer #2: No

3. Have the authors made all data underlying the findings in their manuscript fully available?

Reviewer #1: Yes

Reviewer #2: No

4. Is the manuscript presented in an intelligible fashion and written in standard English?

Reviewer #1: Yes

Reviewer #2: Yes

5. Review Comments to the Author

Reviewer #1: Overall I really enjoyed this manuscript and believe it is a significant contribution towards novel applications of ecological niche models. The authors start from a deep theoretical knowledge and develop their models with the intention of testing a clear hypothesis based on phylogenetic/taxonomic knowledge. This may seem trivial, but it makes this manuscript stand out from other studies that use ENMs to identify past refugia/cradles for species or populations. I believe the strategy employed by the authors to identify cradles is very solid and creative, with a strong basis. However, there are a series of improvements to make it suitable for publication. I will describe the major ones here:

1) Authors need to account for ENM uncertainty in both algorithm and climate scenarios. There are a variety of algorithms and past climate models, in a way that using only one of each is a risky and undesirable strategy. Much of variation in model's results come from those different sources. (see Diniz-Filho et al, 2009 - Partitioning and mapping uncertainties in ensembles of forecasts of species turnover under climate change; Qiao et al 2015 - No silver bullets in correlative ecological niche modelling: insights from testing among many potential algorithms for niche estimation; Varella et al. 2015 - A Short Guide to the Climatic Variables of the Last Glacial Maximum for Biogeographers; Goberville et. al. 2015 - Uncertainties in the projection of species distributions related to general circulation models for discussions about those issues).

There are a series of R packages and scripts that may help you fit ENM using other algorithms (BIOMOD, sdm, ZOON are some examples).

2) While I am satisfied with the strategy of using occurrence data of another species as a proxy of soil in the current period, using those same locations to identify cradles in the past is not straightforward, as the distribution is a interplay of several factors and could be different in the past, which means authors may be missing areas with suitable soil. I recommend the authors to directly include soil variables within the models (see Velazco et al. 2018 - Using worldwide edaphic data to model plant species niches: An assessment at a continental extent; Hengl et al. 2014 - SoilGrids1km — Global Soil Information Based on Automated Mapping) or to also create past models for O.tridentata. There is also the need of a better explanation about the procedures at this step.

3)The authors should explore a little further the effects of their survival threshold. While I understand why the authors chose the LPT (Least Presence Threshold), I believe they should consider using a higher threshold as using the LPT based on data from online

databases is risky due to the data being more prone to errors. The suitability frequencies histogram is a great way to visualize your data, and it seems there is a single point in a value of

very low suitability, the LPT is highly affected by those situations and there is no way to be sure that this point represents a viable population.

Therefore, I recommend exploring other thresholds (a common threshold for ENMs is the value that maximizes specificity and sensitivity, max TPR+TNR) and at least evaluate the effects

of the different threshold in the results.

4)This is a suggestion, and I do not think is an option authors must do for this manuscript. There is a nice R package for plant dynamic plant modelling called MigClim which I believe would be an extremely nice addition to the discussion of the cradle dynamics throughout time.

I made other minor comments and comments related to the text directly in the draft and attached the document.

Again congratulations for your study, it was very pleasant and exciting to read.

Reviewer #2: General comments

The authors did a good job attempting to determine the potential Gypsophila bermejoi cradles, but some assumptions remain unclear. For example, the gypsum soils at Iberian Peninsula are ancient soils? In other words, is it ok to use Ononis tridentata current distribution as a proxy to locate gypsum soils and, ultimately, G. bermejoi LGM cradles? There is scientific evidence to confirm that the studied species don’t occur in another soils unless the gypsum ones?

Considering Methods section, the authors must add information to allow work replication. For example, I could not find the occurrence data. Another important point here is that the authors chose to use only one SDM algorithm (MaxEnt) even though its TSS scores are low, and in the Discussion section they try to explain G. bermejoi historical geographic distribution based only on this result. Actually, the Discussion section seems inconsistent and speculative at some points. I hope the authors improve both sections, Methods and Discussion, once the main paper subject has scientific relevance.

Specific Comments

L 65-68: Funny English.

L 72-74: I can’t see your point here.

L 110: Why these species are of ecological, evolutionary and biochemical interest?

L 111: Which taxon? All of them?

L 115 – 116 and 138 – 146: This is Methods.

Figure 1: The legend do not address step 2. Figures resolution must be improved.

L 144: The word “species” must be excluded.

L 160 – 162: Funny English.

L 175: Occurrence data from GBIF must be cleaned up. Please, make sure that you have avoid bias. A useful option could be the package CoordinateCleaner (see Boria et al. 2014 and Zizka et al. 2019).

L 178: Please, make sure that you have used WorldClim 2 (Fick and Hijmans 2017), avoiding some anomalies of the older version pointed out by Escobar et al. 2014.

L 180: Besides CCSM4, what climate models have you used for past and future projections?

L 184 – 192: What are your predictor variables at first, all the 19 from Bioclim? Which clustering algorithm have you used? Have you performed Jackknife with all the predictor variables or after the correlation dendrogram? Did you include soil traits, since you have mention references 51 and 52? Please, write Methods making sure that anyone can replicate your work.

L 187 and 189: “jack-knife” or “jackknife”?

Table 1: After reading the previous paragraph I have this sensation that you have choose predictor variables according to your personal experience. Please, explain in details how do you get to these four variables.

L 202: The hyperlink is outdated.

L 204: What did you consider “redundant presence”? Which protocol did you use to deal with autocorrelation?

L 209: Couldn’t understand the sentence “The study focused on the effect of climatic oscillations and as such, we used only bioclimatic variables to generate the models.”

L 211: The reference 58 is out of context. If you know for sure that these plants are strictly gypsophyllic and “they are only present on gypsum soils and their suitability on other soils is “0”, please insert a reference that confirms that.

L 214 - 216 : So you adjust the functions to avoid interpreting difficult curves? I don’t understand.

L 217: About model evaluation metrics, I strongly recommend the follow reading to the authors: “Leroy et. al. 2018 - Without quality presence – absence data, discrimination metrics such as TSS can be misleading measures of model performance”.

General comment: There are several algorithms that could be used to perform Species Distribution Modelling / Ecological Niche Modelling, so I want to know why did you choose to use only maxent? Is this the better choice to your data? Why?

L 223: “frequencies would be”

L 271: Where is the Table 2?

L 304: In my understanding you used the current location of Ononis tridentata to identify gypsum soils, assuming that its location is the same since LGM, is that correct? So, could you insert some references which ensures this information?

L 309: Does the algorithm have a reference? Again, make sure that anyone can replicate your work.

Figure 3: Please, make sure that all colors, shadows and dots in your map are explained by the figure’s legend and review all details. The figure is self-explanatory, but with the legend becomes confusing.

L 336: Considering that you are using Species Distribution Modeling and it is a standard way to identify species current distribution (and you can include soils traits as well, see Velazco et al. 2017), why did you use all of these procedures? It seems that you are using current occurrence points to point out where the species.

L 471: You wrote more than three paragraphs without cite anyone besides your previous papers about G. bermejoi. Besides, most of your arguments are speculative, like “According to our hypothesis”, “It seems reasonable to accept”, “We believed”, “We think that”, “Seems difficult to explain”, “We feel that”.

6. PLOS authors have the option to publish the peer review history of their article (what does this mean?). If published, this will include your full peer review and any attached files.

Reviewer #1: Yes: André Felipe Alves de Andrade

Reviewer #2: No

---

## [Author Response · Author response to Decision Letter 0]

14 Feb 2020

Dear all, editors and reviewers

 The following letter gives an answer to all the changes and explanations requested.

Response to the requirements of the journal and reviewers: 

1. We have done again all the models, calculations and maps from scratch. We performed ensembles of models generated using four different algorithms (General Linear Models, MaxEnt, Random Forest and MARS) with 30 bootstrap replications each. 

2. Ensembles using Three General Circulation Models were performed for the projections to the Mid Holocene and Last Glacial Maximum for each species.

3. The Uncertainties were assessed and mapped at the former steps.

4. Gypsum soils maps were used in spite of using the indicator species Ononis tridentata.

5. Several survival tresholds for Gypsophila bermejoi were explored to assess their effects on the results.

6. We have revised all the references suggested and taken them into account to improve our research.

7. All the small comments have been addressed.

to Reviewers 1 and 2 

 We are grateful for the comments made on our study and we appreciate sincerely the encouraging comments. We hope you find all the topics complete and clear with all the information added in this revised manuscript. 

 We appreciate your feedback and hope to have responded effectively.

Kind regards

 The authors

---

## [Decision Letter · Decision Letter 1]

20 Mar 2020

PONE-D-19-14999R1

Using Species Distribution Models to locate the potential cradles of the allopolyploid Gypsophila bermejoi G. López (Caryophyllaceae)

PLOS ONE

Dear Mrs. Bartolomé,

Thank you for submitting your manuscript to PLOS ONE. After careful consideration, we feel that it has merit but does not fully meet PLOS ONE’s publication criteria as it currently stands. Therefore, we invite you to submit a revised version of the manuscript that addresses the points raised during the review process.

We would appreciate receiving your revised manuscript by May 04 2020 11:59PM. To enhance the reproducibility of your results, we recommend that if applicable you deposit your laboratory protocols in protocols.io, where a protocol can be assigned its own identifier (DOI) such that it can be cited independently in the future. For instructions see: http://journals.plos.org/plosone/s/submission-guidelines#loc-laboratory-protocols

We look forward to receiving your revised manuscript.

Kind regards,

Daniel de Paiva Silva, Ph.D.

Academic Editor

PLOS ONE

Additional Editor Comments (if provided):

Dear authors, congratulations! The manuscript is nearly there. Reviewer #1, which was one of the original reviewers believe it is done. Still, the new Reviewer#3, who is also a SDM specialist, still believes very minor corrections that need to be done before the text is ready. As soon as you perform those, please resubmit your manuscript up to June 19th 2020. As soon as I receive it, I will forward it only to reviewer #3, so s/he may evaluate if the minor changes were done.

All the best and beware with the COVID-19 virus.

Reviewers' comments:

Reviewer's Responses to Questions

**Comments to the Author**

1. If the authors have adequately addressed your comments raised in a previous round of review and you feel that this manuscript is now acceptable for publication, you may indicate that here to bypass the “Comments to the Author” section, enter your conflict of interest statement in the “Confidential to Editor” section, and submit your "Accept" recommendation.

Reviewer #1: (No Response)

Reviewer #3: (No Response)

2. Is the manuscript technically sound, and do the data support the conclusions?

Reviewer #1: Yes

Reviewer #3: Yes

3. Has the statistical analysis been performed appropriately and rigorously? 

Reviewer #1: Yes

Reviewer #3: Yes

4. Have the authors made all data underlying the findings in their manuscript fully available?

Reviewer #1: Yes

Reviewer #3: Yes

5. Is the manuscript presented in an intelligible fashion and written in standard English?

Reviewer #1: Yes

Reviewer #3: Yes

6. Review Comments to the Author

Reviewer #1: First of all, congratulations for the time taken to perform the modifications. I know it may not have been simple to run the experiment all over, but it was very well done and I believe the additions really improved the overall quality of the manuscript.

I believe the manuscript is ready to be published and the findings are very interesting!

This is a very nice piece of work and presents a very good application for ecological niche models!

I do believe there are some minor issues the authors must address, but it does not change my recommendation to accept the manuscript.

I found three sentences that I believe are related to the previous version of the manuscript, which state that only Maxent was used, which is not true in this new version. The sentences are at lines 216, 233 and 359. This is only a small issue, but can confuse readers, please address those.

Reviewer #3: Comments to the authors

The study evaluates possible cradles for Gypsophila bermejoi species, an allopolyploid species, based on its parent species G. struthium subsp . struthium and G. tomentosa, both diploids species. For these, they projected the parent species environmental suitability to Last Glacial Maximum (LGM), Mid Holocene (MH), and current time periods, and established as potential cradles regions where both species have low suitability values. Then they compared these cradles areas with the G. bermejoi environmental suitability spatial distributions from LGM, passing through MH, until current time period.

The study is well designed and do have a nice idea and hypothesis. My only concern is about the ensemble approach on SDM analyses. I see that the authors added the ensemble framework after their first revision. Nonetheless, they did not reanalyzed their data according to the ensemble results, maintaining only the MaxEnt results in the construction of cradles hypothesis. I know that is a lot of work to do in reanalyzing all data according to the ensemble results. However, they have to point out that the ensemble SDM analyses was only done to evaluate the uncertainties bringing from different “algorithms” and “AOGCMs”. Moreover, they have to show, in some way, that the results come from MaxEnt do not differ, as much as, the results of the other “algorithms”, somehow analytical. Finally, it was missing if the Maxent results, used to build their hypothesis, was come from the ensemble of the three AOGCMs for LGM, and MH.

Specific comments:

Line 22: “Allopolyploid” is miswritten.

Line 81: “Plans” is miswritten.

Lines 216-219: Why only MaxEnt model? I do not understand this part. And what about the ensemble results?

Line 225: Something is misplaced in this sentence. Please rewrite.

Line 233: I do not understand “The MaxEnt approach Ensembles”. You have to enlighten it in the previous sessions, as I asked in the third comment.

Line 359: Why, again only Maxent? And what about the ensemble results?

7. PLOS authors have the option to publish the peer review history of their article (what does this mean?). If published, this will include your full peer review and any attached files.

Reviewer #1: No

Reviewer #3: No

---

## [Author Response · Author response to Decision Letter 1]

2 Apr 2020

Dear all, editors and reviewers

 The following letter gives an answer to all the changes and explanations requested.

Response to the requirements of the journal and reviewers: 

1. We have corrected all the miswritten words. 

2. Sentences at lines 216, 233 and 359 have been corrected. Those lines mention MaxEnt as the only algorithm used and are remnants of the first version of the manuscript. For the current version all the calculations were repeated from scratch using four algorithms, and three General Circulation Models for the projections to the LGM and Mid Holocene.

to Reviewer 1

Again, we are grateful for your encouraging comments. All the small comments have been addressed, including the corrections of the miswritten words and the sentences at lines 216, 233 and 359.

 We appreciate your feedback and hope to have responded effectively.

to Reviewer 3

We appreciate sincerely the comments made on our study. 

All the mistakes have been corrected, including sentences at lines 216, 233 and 359 which state that MaxEnt was the only algorithm used, which is not true in this new version. Those lines were remnants of the previous version of the manuscript.

In the current version we state clearly that four algorithms were used to perform the ensembles and, later, three General Circulation models were used for the projections to the LGM and Mid Holocene. 

We hope you find all the topics complete and clear with all the information added in this revised manuscript. 

Kind regards

 The authors

---

## [Decision Letter · Decision Letter 2]

22 Apr 2020

Using Species Distribution Models to locate the potential cradles of the allopolyploid Gypsophila bermejoi G. López (Caryophyllaceae)

PONE-D-19-14999R2

Dear Dr. Bartolomé,

We are pleased to inform you that your manuscript has been judged scientifically suitable for publication and will be formally accepted for publication once it complies with all outstanding technical requirements.

With kind regards,

Daniel de Paiva Silva, Ph.D.

Academic Editor

PLOS ONE

Additional Editor Comments (optional):

Congratulations! Well done!

Stay safe in the pandemic!

Reviewers' comments:

Reviewer's Responses to Questions

**Comments to the Author**

1. If the authors have adequately addressed your comments raised in a previous round of review and you feel that this manuscript is now acceptable for publication, you may indicate that here to bypass the “Comments to the Author” section, enter your conflict of interest statement in the “Confidential to Editor” section, and submit your "Accept" recommendation.

Reviewer #3: All comments have been addressed

2. Is the manuscript technically sound, and do the data support the conclusions?

Reviewer #3: (No Response)

3. Has the statistical analysis been performed appropriately and rigorously? 

Reviewer #3: (No Response)

4. Have the authors made all data underlying the findings in their manuscript fully available?

Reviewer #3: (No Response)

5. Is the manuscript presented in an intelligible fashion and written in standard English?

Reviewer #3: (No Response)

6. Review Comments to the Author

Reviewer #3: (No Response)

7. PLOS authors have the option to publish the peer review history of their article (what does this mean?). If published, this will include your full peer review and any attached files.

Reviewer #3: No

---

## [Editor Report · Acceptance letter]

27 Apr 2020

PONE-D-19-14999R2 

Using Species Distribution Models to locate the potential cradles of the allopolyploid *Gypsophila bermejoi* G. López (*Caryophyllaceae*) 

Dear Dr. Bartolomé:

I am pleased to inform you that your manuscript has been deemed suitable for publication in PLOS ONE. Congratulations! Your manuscript is now with our production department. 

With kind regards,

on behalf of

Dr. Daniel de Paiva Silva 

Academic Editor

PLOS ONE